# Genetic Analysis and Epidemiological Impact of SARS-CoV-2: A Multinational Study of 1000 Samples Using RT-PCR

**Talib Banser** [1], **Zainularifeen Abduljaleel** [2,3,4,*], **Kamal H. Alzabeedi** [5], **Adil A. Alzahrani** [6], **Asim Abdulaziz Khogeer** [7,8], **Fadel Hassan Qabbani** [9], **Ahmed T. Almutairi** [10], **Sami Melebari** [2] **and Naiyer Shahzad** [11]

[1] Head of Regional Laboratory of Makkah, Ministry of Health, Makkah 21955, Saudi Arabia
[2] Molecular Diagnostics Unit, Department of Molecular Biology, The Regional Laboratory, Makkah 21955, Saudi Arabia
[3] Science and Technology Unit, Umm Al-Qura University, Makkah 21955, Saudi Arabia
[4] Department of Medical Genetics, Faculty of Medicine, Umm Al-Qura University, Al-Abedia Campus, Makkah 21955, Saudi Arabia
[5] Medical Research and Clinical Biochemistry in Regional Laboratory of Makkah, Ministry of Health, Makkah 21955, Saudi Arabia
[6] Quality Management in Regional Laboratory of Makkah, Ministry of Health, Makkah 21955, Saudi Arabia
[7] Research Department, The Strategic Planning, General Directorate of Health Affairs Makkah Region, Ministry of Health (MOH), Makkah 21955, Saudi Arabia
[8] Medical Genetics Unit, Maternity & Children Hospital, Makkah Healthcare Cluster, Ministry of Health (MOH), Makkah 21955, Saudi Arabia
[9] Technicians Supervisor in Regional Laboratory of Makkah, Ministry of Health, Makkah 21955, Saudi Arabia
[10] TB Department in Regional Laboratory of Makkah, Ministry of Health, Makkah 21955, Saudi Arabia
[11] Department of Pharmacology and Toxicology, Faculty of Medicine, Umm Al-Qura University, Makkah 21955, Saudi Arabia
[*] Correspondence: zaabduljaleel@uqu.edu.sa; Tel.: +966-570552361

**Abstract:** The ongoing global public health challenge posed by the COVID-19 pandemic necessitates continuous research and surveillance efforts. In this study, we comprehensively analyzed over 1000 COVID-19 RT-PCR tests conducted on a cohort of 1200 patients in Saudi Arabia. Our primary goal was to investigate mutations in specific genes RdRp, N, and E different infection and recovery stages in Saudi patients with SARS-CoV-2. We also extended our analysis to include patients of various nationalities residing in Saudi Arabia, with the overarching objective of assessing these genes as markers for COVID-19 presence and progression. To diagnose and investigate potential genetic variations in COVID-19, we engaged RT-PCR. Our study primarily focused on detecting mutations in the RdRp, N, and E genes in Saudi patients with SARS-CoV-2, as well as individuals from various national residing in Saudi Arabia. This molecular technique provided valuable insights into the virus's genetic makeup during infection and recovery. In our analysis of 671 positive COVID-19 cases, diverse gene involvement patterns were observed. Specifically, 55.91% had mutations in all three genes (RdRp, N, and E), 62.33% in both N and E genes, and 67.16% in RdRp and N genes. Additionally, 30.75% exhibited mutations exclusively in the RdRp gene, and 51.58% had mutations in the N gene. The N gene, in particular, showed high sensitivity as a marker for identifying active viral circulation. Regarding the temporal dynamics of the disease, the median duration between a positive and a subsequent negative COVID-19 RT-PCR test result was approximately 33.86 days for 44% of cases, 14.31 days for 30%, and 22.67 days for 4%. The insights from this study hold significant implications for managing COVID-19 patients during the ongoing pandemic. The N gene shows promise as a marker for detecting active viral circulation, potentially improving patient care and containment strategies. Establishing a defined positive threshold for diagnostic methods and correlating it with a low risk of infection remains a challenge. Further research is needed to address these complexities and enhance our understanding of COVID-19 epidemiology and diagnostics.

**Keywords:** COVID-19; coronavirus disease; SARS-CoV-2; RT-PCR; pandemic

## 1. Introduction

The emergence of the 2019 coronavirus pandemic, driven by severe acute respiratory syndrome coronavirus 2 (SARS-CoV-2), has positioned itself as a paramount global concern, a fact underscored by the World Health Organization's declaration [1]. Originating in late 2019, the initial cases of individuals presenting severe respiratory infections and pneumonia were identified within China's Hubei Province [2,3]. The rapid and extensive transmission of this highly contagious virus sent shockwaves across the global community, impacting nearly every corner of the world. By the first half of 2020, the number of confirmed COVID-19 cases worldwide had surged beyond seven million, carrying with it the heavy burden of over 400,000 associated fatalities [4,5]. COVID-19, the disease caused by the SARS-CoV-2 virus, manifests a range of common symptoms, such as fever, coughing, shortness of breath, and fatigue [3,6]. What is notably remarkable is the estimated case fatality rate (CFR) for COVID-19, which hovers between 3.4% and 6.6%, a rate notably lower when compared to previous outbreaks like severe acute respiratory syndrome (SARS) with a CFR of 9.6% and Middle East respiratory syndrome (MERS) with a staggering CFR of 34.3% [7].

The epidemiological landscape of COVID-19 was marked by a pivotal moment when, on 4 February 2020, data pointed to the primary reproductive number (R0) standing at 2.2 at the commencement of the epidemic (30 January 2020) until 1 February 2020, with a range extending from 3.6 to 5.8. This was coupled with the estimated epidemic doubling time in Wuhan, China, which was measured at 3.6 days, fluctuating between 1.0 and 7.7 days [8,9]. The declaration of COVID-19 as a pandemic by the World Health Organization (WHO) on 11 March 2020 marked a crucial juncture. The pandemic's extraordinary contagiousness, coupled with the absence of pre-existing immunity, facilitated its relentless propagation. The reproductive number (R0) of COVID-19 surpassed that of influenza, necessitating governments across the globe to institute stringent measures, including lockdowns, in an effort to curtail human movement and social interactions, effectively dampening the trajectory of the epidemic.

Subsequent to the implementation of lockdowns, a pivotal transformation occurred in our understanding of the COVID-19 pandemic. Data emerging from the initial outbreak in Wuhan, China, revealed a noticeable reduction in the time-varying reproductive number (Rt) during February 2020 [10]. This decline in Rt strongly indicated that the measures instituted through lockdowns and restrictions were indeed effective in curbing the rapid spread of the virus.

However, it is essential to highlight that the battle against COVID-19 was not solely fought on the epidemiological front. An equally significant development was the early access to the complete genome of SARS-CoV-2, the virus responsible for COVID-19. This scientific milestone paved the way for the development of highly specific primers and the standardization of laboratory protocols essential for COVID-19 diagnosis. Notably, one such diagnostic method, the real-time reverse transcription polymerase chain reaction (RT-PCR) assay, was meticulously crafted to target specific genetic regions of SARS-CoV-2. This assay, focusing on the RNA-dependent RNA polymerase (RdRp), envelope (E), and nucleocapsid (N) genes of the virus, was published on 23 January 2020 [11,12]. Among these genetic targets, the RdRp assay stood out due to its remarkable analytical sensitivity, offering significant promise for the precise diagnosis and continuous monitoring of COVID-19 cases. However, amidst these remarkable strides in diagnostic methods and molecular virology, there remained a notable gap in our understanding. A comprehensive investigation into the expression levels of the RdRp, N, and E genes within SARS-CoV-2 patient specimens had been conspicuously absent. It is this knowledge gap that served as the impetus for the present study.

Our research is not confined to the conventional boundaries of molecular virology or epidemiology. Instead, it embarks on a profound exploration by conducting a meticulous big data analysis of the global epidemiological landscape of SARS-CoV-2. Through this journey, we delve deep into the dynamic expression profiles of the RdRp, N, and E

genes within patients, tracing their trajectory from diagnosis to their role in shaping the broader pandemic response. The use of rigorous statistical analyses lies at the heart of our methodology, aimed at unearthing nuanced insights concealed within these pivotal genetic markers.

In essence, our study transcends the typical confines of conventional research by offering a comprehensive view of the COVID-19 pandemic. It is our hope that this undertaking will not only fill a crucial void in the understanding of SARS-CoV-2 but also provide vital insights for policymakers, healthcare professionals, and researchers seeking to navigate the complex terrain of this global health crisis. This comprehensive study, therefore, endeavors to unravel the multifaceted nature of the COVID-19 pandemic by shedding light on its epidemiological dynamics and the genetic aspects of the virus. Through this, it contributes to the collective understanding of the disease and provides insights that can potentially guide future research and public health strategies.

## 2. Materials and Methods

### 2.1. Patient Sample Collection and Dataset

Upon the arrival of samples from various Saudi hospitals, a rigorous and standardized SARS-CoV-2 testing protocol was employed to ensure the accuracy and reliability of the diagnostic process. The study cohort consisted of a total of 1200 patients who were admitted to healthcare facilities between the months of January and May in the year 2020. To comprehensively detect the presence of the SARS-CoV-2 virus, the real-time reverse transcription polymerase chain reaction (RT-PCR) method was employed, targeting all three essential genes of the virus, namely RdRp, E, and N. In line with the protocols established by the National Center for Infectious and Parasitic Diseases, a comprehensive dataset was meticulously curated, encompassing a wide array of information for each patient. This dataset included essential details such as patient names, contact information, date of sample collection, presenting symptoms, existing comorbidities, epidemiological information, the onset of symptoms, and demographic characteristics. All data collection processes adhered to the guidelines set forth by the National Center for Infectious and Parasitic Diseases, ensuring consistency and uniformity in information capture (as illustrated in Figure 1).

To further categorize and classify patients based on clinical criteria, our study aligned with the interim guidance provided by the World Health Organization (WHO) [13]. Patients who exhibited pneumonia-like symptoms, irrespective of whether they presented with imaging findings, were classified as having symptoms consistent with COVID-19. This approach was in accordance with international standards for identifying potential cases of the disease. One of the key strengths of our study lies in the utilization of the entire cohort of patient samples. This extensive dataset allowed us not only to diagnose and analyze COVID-19 cases but also to extrapolate the prevalence of SARS-CoV-2 within the broader Saudi Arabian population. This epidemiological perspective was instrumental in determining the appropriate levels of statistical significance needed to discern and interpret the observed differences within our dataset. By undertaking this approach, we aimed to contribute to a more holistic understanding of the virus's impact and epidemiological dynamics in the Saudi Arabian context.

The sample size ($n$) were calculated according to the formula:

$$n = \frac{2(1-p)pz^2}{e\left(\frac{(1-p)pz^2}{2eN} + 1\right)}$$

The equation for calculating the margin of error ($e$) for a given proportion ($p$), population size ($N$), and confidence level ($\alpha$) of 95% is $e = z \times \text{sqrt}(p \times (1-p)/N)$. Here, $z = 1.96$.

$$z = 1.96, p = 0.5, N = 35{,}000{,}000, e = 0.05$$

$$n = \frac{1.962 \cdot 0.5 \cdot \frac{1 - 0.5}{0.052}}{1 + \left(1.962 \cdot 0.5 \cdot \frac{1 - 0.5}{0.052 \cdot 35{,}000{,}000}\right)}$$

$$n = 384.16/1 = 384.156$$

$$n \approx 385 \text{ patients}$$

The sample size with a finite population correction was calculated to be 385.

## COVID-19 Diagnostic Test through RT-PCR

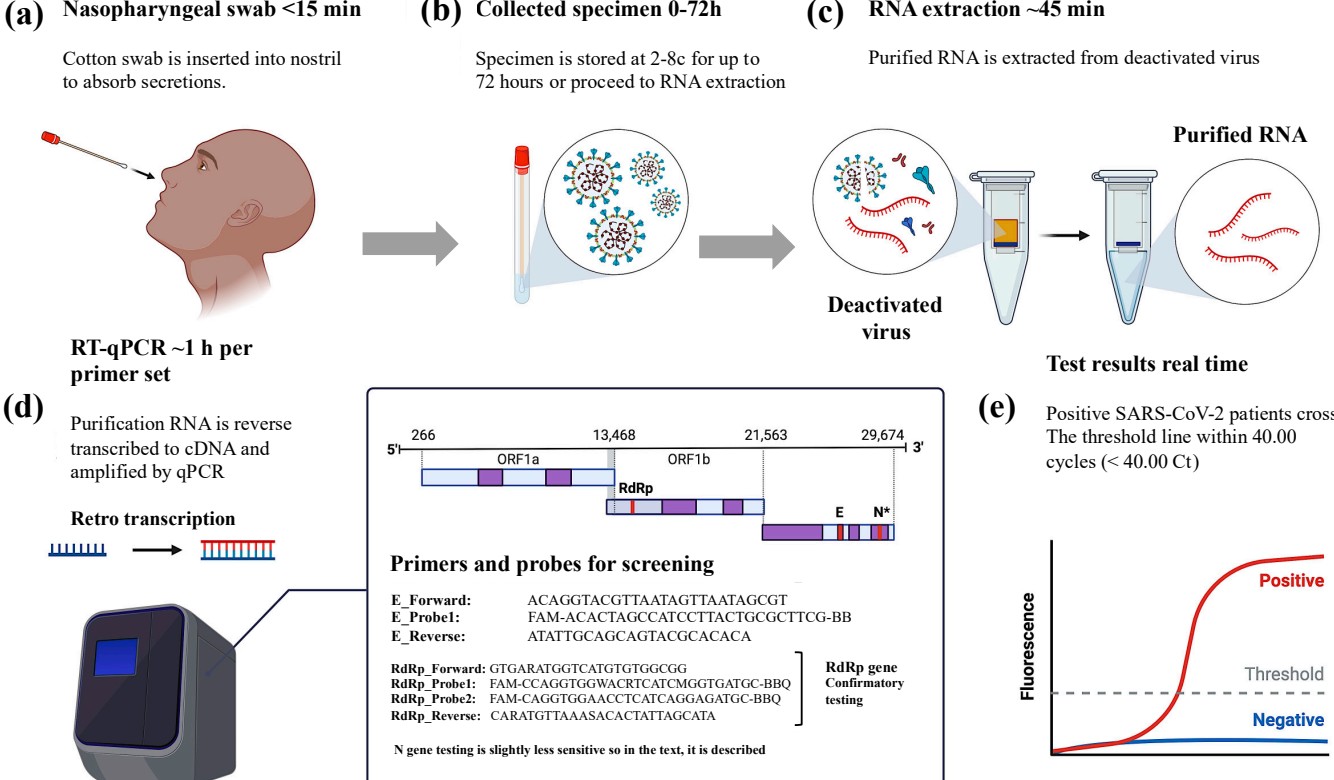

**Figure 1.** Massive and rapid COVID-19 testing by extraction of SARS-CoV-2 RT-PCR.

### 2.2. Design of qRT-PCR Primers Targeting E, N, and RdRp Genes

We employed the proposed primer pair to PCR amplify the target genes E, N, and RdRp using a CLC workbench. To verify the primer pair's coverage and alignment with the target sequences, we conducted a nucleotide base blast on all three-primer sequences. This involved importing them as separate sequences and assessing primer-binding locations, considering scores and mismatches at the primer 3′ end [14].

### 2.3. Detection of SARS-CoV-2 through RNA Isolation and RT-PCR

In our laboratory, we conducted the extraction of total nucleic acid (TNA) from both clinical specimens and viral isolates using the NucliSENS easyMAG system, which is manufactured by bioMérieux located in Marcy-l'Étoile, France. The specific elution volume used for this extraction process was determined based on the type of specimen, with a volume of 55 μL employed for respiratory tract specimens, urine, rectal swabs, and feces. For plasma specimens, a larger elution volume of 100 μL was utilized, while a volume of 25 μL was used for extractions. All extracted TNA samples were subsequently stored at a temperature of −80 °C and were prepared for use in all reverse transcription polymerase chain reaction (RT-PCR) tests. The actual process of detecting and analyzing

the SARS-CoV-2 genes was carried out using the Bio-Rad CFX96 Real-time PCR system, identified by the reference number 1854095-IVD. This system allowed us to examine the amplification curves and Ct values within 45 cycles, which are crucial parameters for identifying the presence of the virus in the samples. In addition to the SARS-CoV-2 genes, we included an internal control amplification step in our testing procedures. This control step was essential for confirming the quality of the RNA extracted from the specimens and for ensuring the reliability and accuracy of our RT-PCR tests.

### 2.4. Statistical Analysis

We employed descriptive statistics to analyze the various variables under consideration, including measures such as means, medians, standard deviations, and interquartile ranges. Variables with distinct characteristics were represented using whole numbers and percentages. To assess potential associations between categorical variables and to compare the performance of our study, we utilized cross-tabulation, the Chi-squared test, and Fisher's exact test. Statistical significance was determined when the *p*-value was less than 0.05. The data analysis was conducted using GraphPad Software, Inc.'s software, specifically version 9.3.1 (La Jolla, CA, USA).

In order to determine the mean through simulation, we conducted an investigation into the properties of a confidence interval (CI) for "R" and "Shiny". This exploration considered how the confidence level and the shape of the population distribution influenced the properties of the confidence interval. Categorical data were presented with corresponding 95% confidence intervals (95% CIs) (Figure 2). To calculate sensitivity, specificity, positive predictive value (PPV), and negative predictive value (NPV), we employed an online statistical tool designed for NPV calculations.

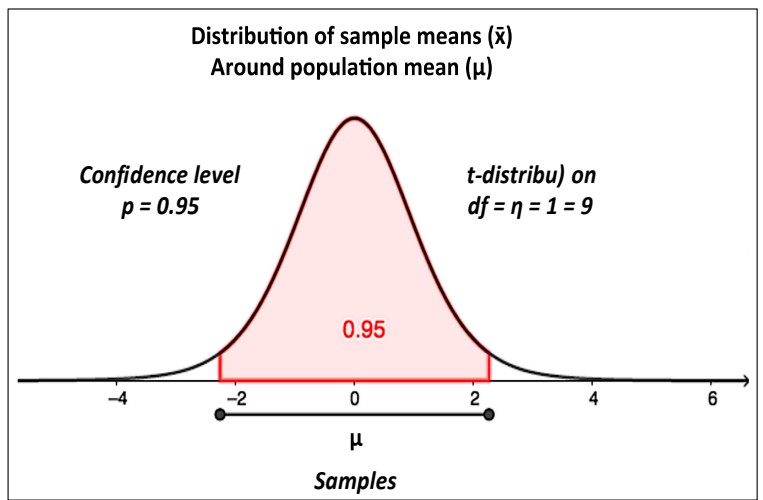

**Figure 2.** Means and distributions of standard deviation.

Furthermore, we examined the effect of increasing sample size on the properties of confidence intervals. Using a sample mean and simulation, we computed the confidence interval and margin of error when multiple samples were drawn from the same population distribution. The sample mean varied within the confidence interval $\bar{x} \pm E$, encompassing the population mean (μ) in 95% of the samples.

Confidence interval: $35{,}070 \pm 5.000$ ($\pm 0.01\%$) [35,065.000–35,075.000]

$$\bar{x} \pm Z \frac{s}{\sqrt{n}}$$

Assuming a 95% confidence level, the following variables can be denoted: $Z$ as the $Z$-value for the specific confidence level, $\overline{x}$ as the sample mean, $\sigma$ as the standard deviation, and $n$ as the sample size.

$$
\begin{aligned}
\text{Cl} \quad &= \overline{X} \pm Z \times \frac{s}{\sqrt{n}} \\
&= 35{,}070 \pm 1.9600 \times \frac{15{,}092.32}{\sqrt{35{,}000{,}000}} \\
&= 35{,}070 \pm 5.000
\end{aligned}
$$

The variables $Z$, $\overline{x}$, $\sigma$, and $n$, respectively, represent the confidence level, sample mean, standard deviation, and sample size. To compute the confidence intervals and margins of error with a 95% confidence level, the sample mean can be used as it often follows a normal distribution.

## 3. Results

### 3.1. Effective Primers Designed and PCR Validation

We developed a set of highly versatile primers designed for the detection and quantification of gene expression. These primers were crafted using gene sequences obtained from the NCBI database, and the specifics of these primers, generated through dedicated software, are presented in detail in Table 1. The software provided valuable insights into various essential properties, including melting temperatures, GC content, and primer efficiency. Several properties are critical for ensuring the success of PCR reactions, and they encompass aspects like primer length, GC percentage, efficiency, and melting temperature (Tm). According to our dataset, the proposed primers were designed to be 20 base pairs in length, highly specific, and demonstrated effective binding capabilities to their respective target sequences. Notably, the primers designed for the E, N, and RdRp genes exhibited GC contents of 52.9%, 55%, and 59%, and melting temperatures of 59.61 °C, 60.04 °C, and 97.55 °C, respectively (as depicted in Figure 3a–c). These melting temperatures fall within the typical range considered suitable for RT-PCR reactions, ensuring optimal conditions for gene amplification. In the primer design process, it is also crucial to account for the temperature at which secondary structures may form, as this can impact the outcomes of PCR experiments. Based on our results, these primers exhibit the potential to yield successful PCR results while satisfying essential design criteria.

**Table 1.** Primer sequences are selected based on analysis of RT-PCR data.

| Gene | Sequence (5′->3′) | Template Strand | Length | Start | Stop | Tm | GC% |
|---|---|---|---|---|---|---|---|
| Envelop (E) | ACTACTCTGGTGTGTGGTGC | Plus | 20 | 45 | 64 | 59.61 | 55 |
| | ACTCGTTTAGGGAAAGGGTCT | Minus | 21 | 221 | 201 | 58.38 | 47.62 |
| Nucleocapsid (N) | ACAGGTTACGGTGTTAGGCG | Plus | 20 | 337 | 356 | 60.04 | 55 |
| | AGGAGTACCCGTTTTCGCTG | Minus | 20 | 633 | 614 | 60.04 | 55 |
| RNA-dependent RNA polymerase (RdRp) | CACCTACACACCTCAGCGTT | Plus | 20 | 17,773 | 17,792 | 59.97 | 55 |
| | GCACGAACGTGACGAATAGC | Minus | 20 | 17,959 | 17,940 | 59.98 | 55 |

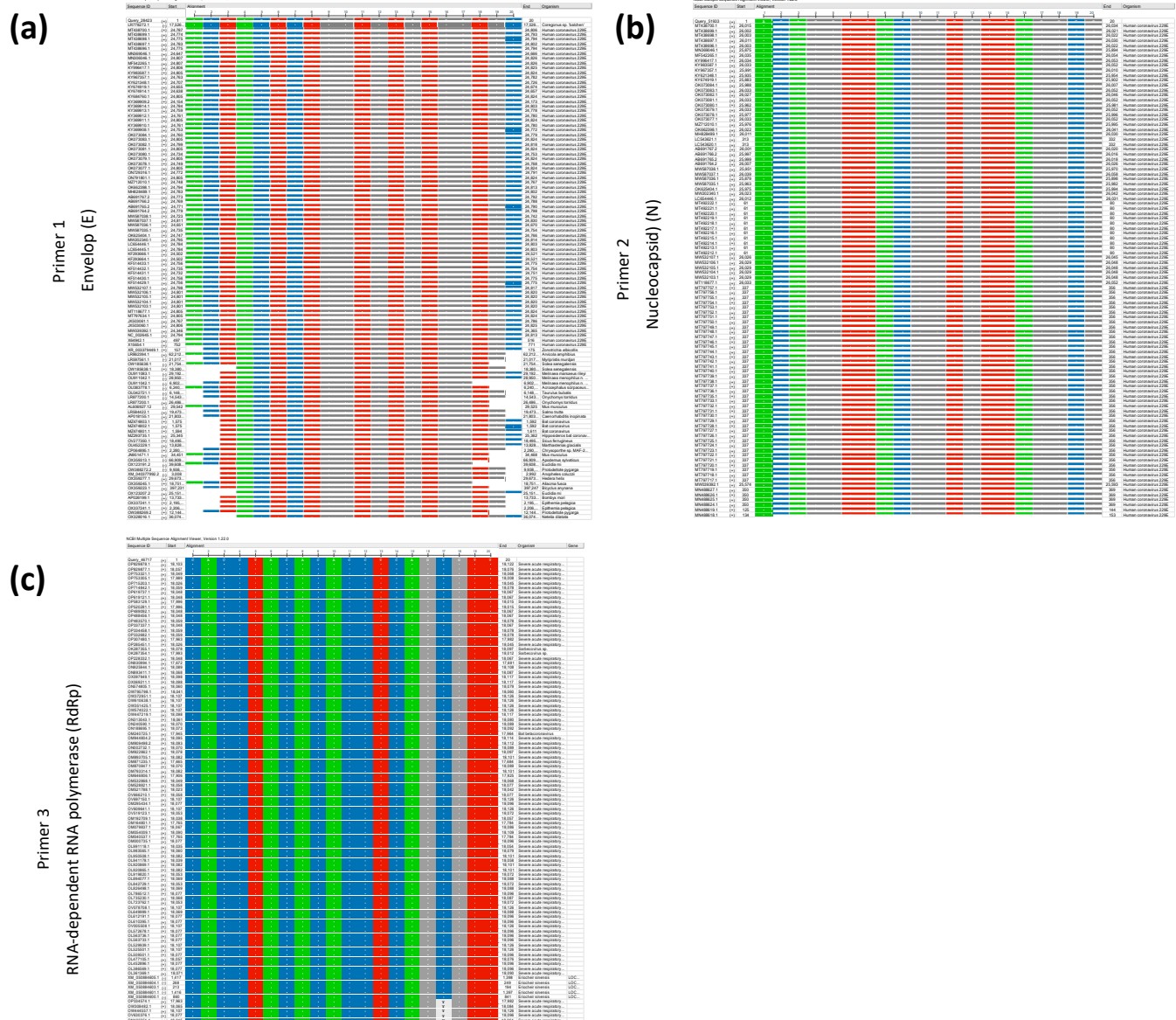

**Figure 3.** (**a**) The nucleotide sequences of primers and probes are compared to primer-BLAST for the envelope gene (E). (**b**) The primer-BLAST program validates the nucleotide sequence of nucleocapsid (N) primers and probes. (**c**) The RNA-dependent RNA polymerase gene (RdRp) primer and probe nucleotide sequences are compared with primer-BLAST to validate the primers and probes.

### 3.2. Wide-Ranging Characteristics of Patient Sample Distribution

The SARS-CoV-2 genome was detected using real-time RT-PCR, with a focus on the E, RdRp, and N genes. Simplex qRT-PCR assays were employed to evaluate all primers and probes. Careful calibration of the qRT-PCR equipment was conducted before preparing the reactions to ensure optimal fluorescence signal generation. Simplex reactions targeting the virus's E, N, and RdRp genes, along with internal control genes (HPRT and RP), were carried out in triplicate.

As per the COVID-19 diagnosis criteria, a Ct value of 37.00 indicated a positive sample, a Ct value of 40.00 signified a negative sample, and a Ct value of 40.00 was indicative of a suspect sample. The total number of COVID-19 RT-PCR assays conducted on the 1000 COVID-19 patients included in this study was 988, and their frequency and distribution are depicted in Figures 4 and 5. The mean age of the patients was 33.86 years, with a standard deviation of 0.499. The study cohort comprised 68.3% male and 31.7%

female patients. Of the 988 patients, 317 (32%) received negative results, while 671 (67.91%) tested positive in the COVID-19 RT-PCR tests (as shown in Table 2). Among the patients with negative results, 221 (69.71%) were male, and 96 (30.28%) were female.

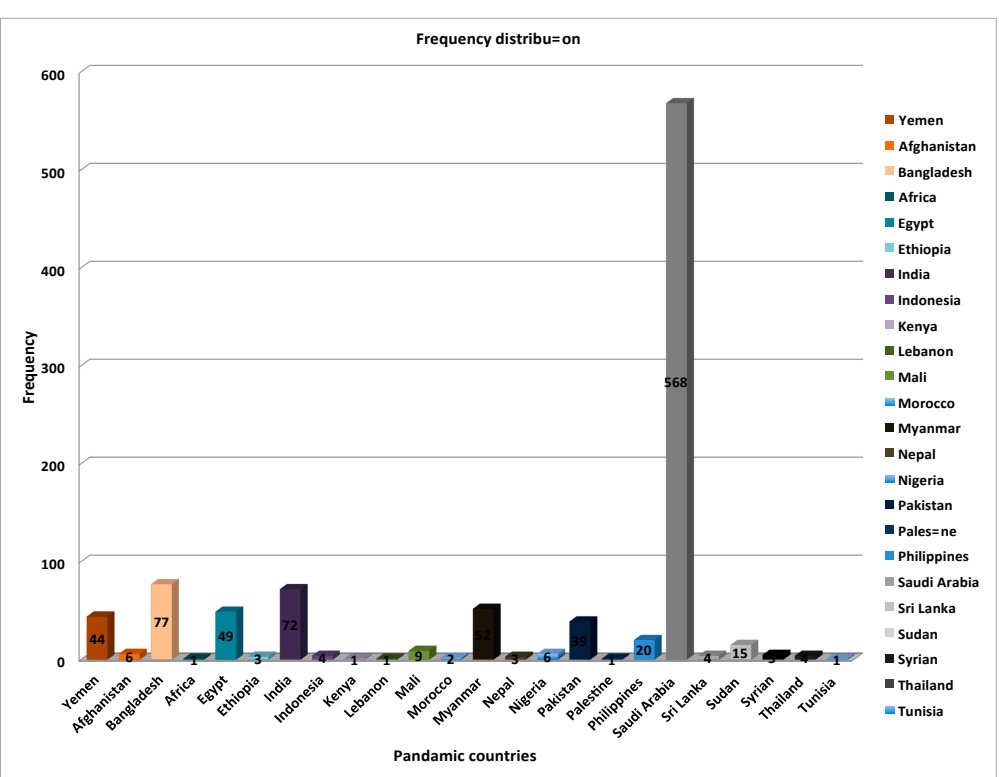

**Figure 4.** Different countries affected by the COVID-19 pandemic are grouped according to their frequency.

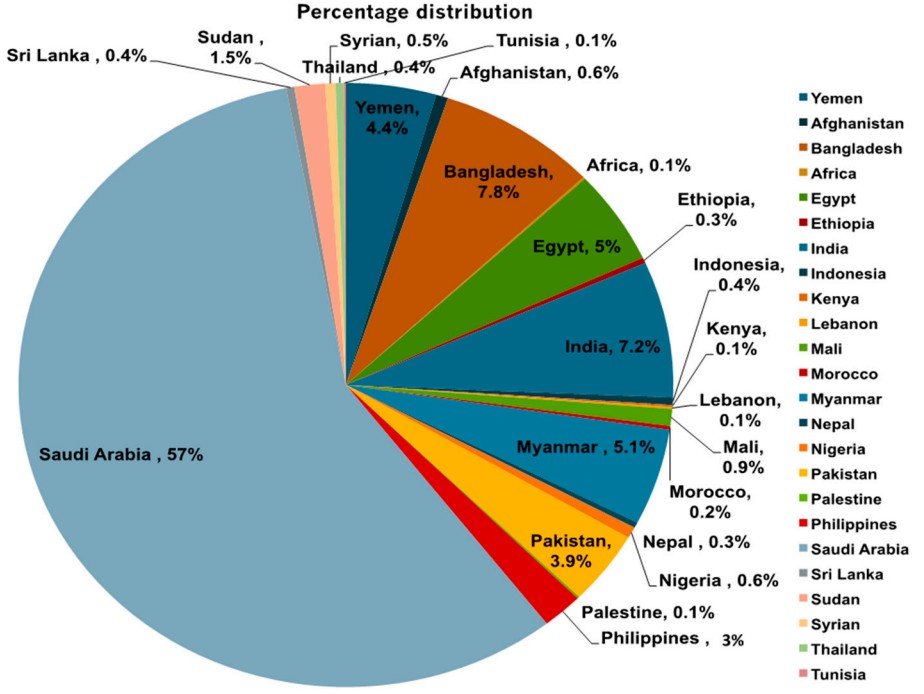

**Figure 5.** Comparison of the percentage distribution of COVID-19 pandemic countries with gendered impacts.

**Table 2.** The frequency distribution of the three targeted genes across different categories.

| Measures | Groups | Frequency | Percent Frequency (%) |
|---|---|---|---|
| Age | | $33.86 \pm 0.499$ | |
| Age groups | >60 years | 931 | 94.2 |
| | ≤60 years | 57 | 5.8 |
| Gender | Female | 313 | 31.7 |
| | Male | 675 | 68.3 |
| E gene | Negative | 671 | 67.9 |
| | Positive | 317 | 32.1 |
| N gene | Negative | 551 | 55.8 |
| | Positive | 437 | 44.2 |
| RdRp gene | Negative | 619 | 62.7 |
| | Positive | 369 | 37.3 |

*3.3. SARS-CoV-2 Genes Detected via Quantitative RT-PCR*

In the majority of cases, the E gene (454 cases, 67.25%), N gene (374 cases, 55.40%), and RdRp (416 cases, 61.62%) were responsible for positive findings. Among the positive cases, females accounted for 217 (69.32%) in the E gene, 177 (56.54%) in the N gene, and 203 (64.85%) in the RdRp gene. The gender distribution in COVID-19 positive cases showed that 31.7% were female, while 68.3% were male, indicating a distinct reaction of the primers and probes toward COVID-19 genes based on gender (as shown in Table 3). Of the total 988 samples, which included both male and female subjects, those aged over 60 had 636 positive cases and 295 negative cases for the E gene. For the N gene, both age groups had positive and negative results, with 519 positive cases and 412 negative cases for those over 60. As for the RdRp gene, only the over-60 group exhibited a high positivity rate, with 586 positive cases and 345 negative cases, while the younger age group had 33 positive cases and 24 negative cases. In the end, reports of COVID-19 infections were classified as "inconclusive", and the total of 988 samples included both positive and negative cases.

**Table 3.** The three genes associated with gender and age groups.

| Measures | Groups | E Gene | | χ2 | P (1-Sided) | OR | 95% CI | |
|---|---|---|---|---|---|---|---|---|
| | | Positive | Negative | | | | LOWER | UPPER |
| Gender | Female | 217 | 96 | 0.420 | 0.283 | 1.100 | 0.824 | 1.469 |
| | Male | 454 | 221 | | | | | |
| Age Groups | >60 | 636 | 295 | 1.177 | 0.173 | 1.355 | 0.781 | 2.351 |
| | ≤60 | 35 | 22 | | | | | |
| | | **N Gene** | | | | | | |
| | | Positive | Negative | | | | | |
| Gender | Female | 177 | 136 | 0.113 | 0.395 | 1.047 | 0.799 | 1.372 |
| | Male | 374 | 301 | | | | | |
| Age Groups | >60 | 519 | 412 | 0.003 | 0.533 | 0.984 | 0.574 | 1.687 |
| | ≤60 | 32 | 25 | | | | | |
| | | **RdRp Gene** | | | | | | |
| | | Positive | Negative | | | | | |
| Gender | Female | 203 | 110 | 0.951 | 0.183 | 1.149 | 0.869 | 1.519 |
| | Male | 416 | 259 | | | | | |
| Age Groups | >60 | 586 | 345 | 0.585 | 0.264 | 1.235 | 0.718 | 2.125 |
| | ≤60 | 33 | 24 | | | | | |

Categorical data analysis yielded a normal positive predictive value (PPV) of 100% and a normal negative predictive value (NPV) of 100%. However, the confidence intervals (CIs) for PPV were at 90.3%, while the NPV had a 95% CI, indicating a 95% CI accuracy. Utilizing Statcrunch, it was evident that the interval had approximately 90% confidence. By establishing a real 95% confidence level with a line corresponding to 95% of the population mean (Figure 6a,b), each green horizontal line represented a confidence interval, with the black vertical line indicating the true population mean. With the sample size approaching 988, the width of the confidence interval decreased significantly. As depicted in Figure 4,

the 95% confidence intervals for 100 samples of size 10 from a Gaussian distribution with an actual mean of 10 were illustrated. Out of 100 intervals, 94 captured the true value of 10, meaning that 95 out of 100 confidence intervals would capture the true population parameter due to the effects of sampling variation in a random set of 100 intervals.

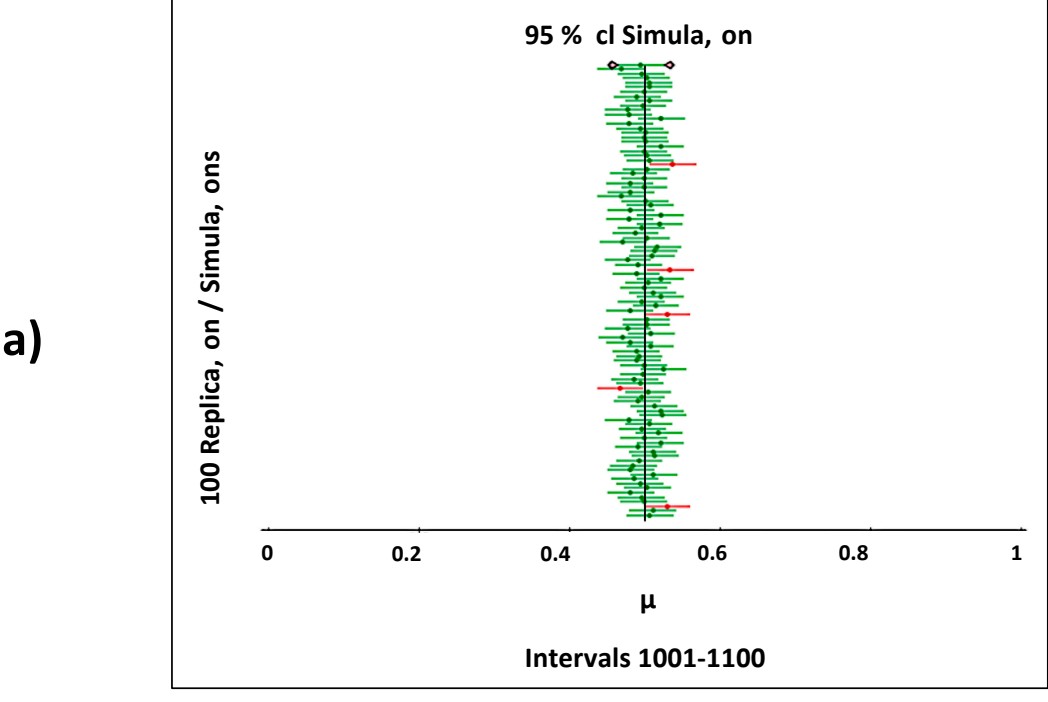

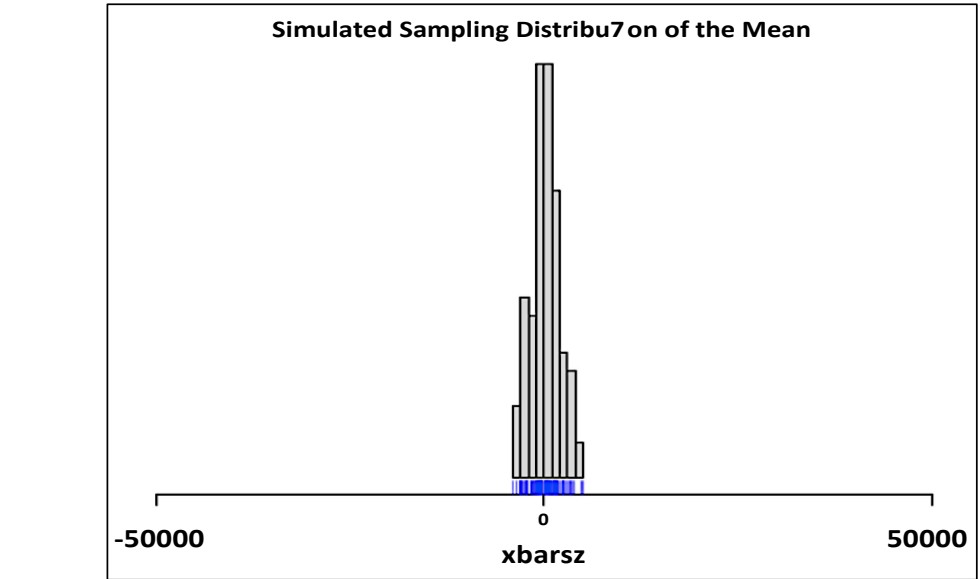

**Figure 6.** (**a**) According to the statistical analysis software StatCrunch [15–19], we were able to calculate 1000 90% confidence intervals based on a population proportion of *p* = 0.22 and a sample size of 988 from Saudi Arabia. The default confidence level of 0.90 was used for the ER Data StatGraph applets, with a default proper population proportion of 0.50. We conducted 100 simulation runs using these default values, and the green and red lines in the "Prop" section represent the value of a prop using the black vertical line definition and the confidence intervals with an actual population proportion of 0.50 between their lower and upper bounds. We also performed an additional 1000 simulation runs and calculated 1000 confidence intervals using the "1000 intervals" table. The "Prop" field indicates 95% confidence in a single computed confidence interval assuming a "Prop" value of 90% (=0.90). In (**b**), Xbarsz is an estimator of μ, while s (sample standard deviation) is an estimator of σ.

## 4. Discussions

Real-time RT-PCR, often referred to as the "gold standard", has emerged as a cornerstone in the global response to the COVID-19 pandemic. This method is instrumental in detecting viral RNA in clinical samples, including sputum, nasal swabs, and throat swabs [20,21]. As the number of COVID-19 cases surged rapidly, the accurate detection of the virus became paramount for effective outbreak control. However, it is important to recognize that real-time RT-PCR results can be influenced by various factors, encompassing the choice of the primer/probe set, viral load, sample quality, anatomical site for sample collection (e.g., mouth or nasal cavity), and timing of sampling [22]. Notably, the sensitivity, specificity, and efficiency of the numerous available primer and probe sets can exhibit substantial variations, contributing to the ongoing challenge of result consistency.

In addressing this issue, certain key genes for SARS-CoV-2 screening, namely E, N, and RdRp, have demonstrated exceptional accuracy and have garnered endorsement from globally recognized health authorities such as the World Health Organization (WHO) and the Centers for Disease Control and Prevention (CDC) [23–25]. It is important to underscore that even within the framework of RT-PCR, the Ct values for positive samples may vary across these gene targets (E, N, RdRp). This variation holds clinical significance, as it underscores the point that while RT-PCR is widely accepted as the standard for COVID-19 testing, not all positive results, especially those with high Ct values, necessarily equate to active infectivity. This complexity underscores the need for a nuanced interpretation of results, considering factors beyond mere positivity to assess infectivity and inform public health measures effectively.

In March 2020, the Ministry of Health (MOH) in the Kingdom of Saudi Arabia (KSA) reported its first COVID-19 case. By December 2020, Saudi Arabia had documented 358,713 cases and 5965 fatalities, as confirmed by the MOH's official website [26–28]. This study was primarily centered on the Saudi Arabian population, involving the analysis of 988 samples from COVID-19 RT-PCR tests. These samples were sourced from 57% of COVID-19 patients at the Makkah regional laboratory. It is worth noting that this research represents an initial step in our broader exploration of SARS-CoV-2-infected patients, and our future studies will delve deeper into this analysis. It is crucial to underscore that comprehending the pandemic's impact in KSA and other countries hinges significantly on our understanding of the clinical and epidemiological factors related to COVID-19 mortality. This research forms a vital part of the collective effort to enhance this understanding, ultimately contributing to more effective strategies for pandemic control and management.

Our study offers valuable insights into the duration of virus shedding in lower respiratory tract samples. We observed that in 55% of cases, shedding persisted for more than 14 days, reaching its peak approximately three weeks after the onset of symptoms. The accurate determination of shedding duration is of paramount importance for effective epidemic control and prevention. The availability of a highly sensitive and specific diagnostic assay is indispensable for the identification of cases, efficient contact tracing, and the diagnosis of suspected SARS-CoV-2 infections. Our findings indicate that in respiratory samples, shedding endured for approximately 18–20 days, with the highest shedding levels occurring in the respiratory specimens of positive cases around 10 to 14 days after the onset of symptoms [29–35]. These insights have significant implications for public health strategies aimed at curbing the spread of the virus.

On a global scale, COVID-19 has had a more significant impact on individuals from countries other than Saudi Arabia, particularly those from India (7.2%) and Bangladesh (7.7%). Our study revealed a higher susceptibility to infection among males, with 68.33% of cases being male, while 31.68% were female, consistent with findings from previous research [24]. We also identified a substantial difference in infection rates between different age groups (>60 and ≤60), with a calculated $p$-value of 0.009 and a confidence interval (CI) of 1.30. This observation aligns with other studies that have reported an elevated risk of SARS-CoV-2 infection among older individuals. This higher risk may be attributed to the virus's ability to transmit in the alveoli, which contain higher levels of angiotensin-

converting enzyme 2 [25–28]. These findings underscore the importance of tailoring public health measures to different demographics, especially as we work to mitigate the impact of the pandemic.

## 5. Conclusions

This comprehensive study has illuminated critical aspects of COVID-19 detection and its far-reaching epidemiological implications. It underscores the pivotal role of real-time RT-PCR as the foundation of COVID-19 diagnostics, while delving into the behavior of key viral genes, namely E, N, and RdRp. The global response to the pandemic has consistently emphasized the importance of precise diagnostics, and our research offers invaluable insights into this aspect. We have highlighted that, even though real-time RT-PCR is considered the gold standard, its reliability is contingent on several factors. Hence, there is a pressing need for the development of highly specific and sensitive assays to augment its performance.

Our findings provide profound insights into the dynamics of the pandemic in Saudi Arabia, while also underscoring the need for comprehensive studies that encompass the influence of clinical and epidemiological factors. The duration of virus shedding in respiratory samples and the observed variations in susceptibility among different demographics serve as stark reminders of the multifaceted nature of COVID-19. These revelations reinforce the necessity for tailored public health measures and underscore the ongoing need for research and vigilance. This study, therefore, makes a substantial contribution to our understanding of COVID-19 and will play a vital role in refining containment strategies, diagnostic methods, and the formulation of public health policies designed to mitigate the impact of the ongoing pandemic.

**Author Contributions:** Conceptualization, T.B.; methodology and software, Z.A.; validation, K.H.A.; formal analysis, A.A.A.; investigation, A.A.K.; visualization, F.H.Q. and A.T.A.; supervision, S.M.; Writing—Reviewing and Editing, T.B. and N.S. All authors have read and agreed to the published version of the manuscript.

**Funding:** This research received no external funding.

**Institutional Review Board Statement:** The research was conducted in compliance with the ethical guidelines established by the Ministry of Health and received approval on 31 January 2021 (Approval Reference Number: IRB No: H-02-K-076-1220-430).

**Informed Consent Statement:** Prior to their participation in the study, informed consent was obtained from all subjects, ensuring they were fully informed about the research and voluntarily agreed to take part.

**Data Availability Statement:** The data supporting the findings of this study is available upon request.

**Conflicts of Interest:** The authors declare no conflict of interest.

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
