# Peer review of "Genetic Analysis and Epidemiological Impact of SARS-CoV-2: A Multinational Study of 1000 Samples Using RT-PCR"

_2673-8007, doi:10.3390/applmicrobiol4010010_

Round 1

Reviewer 1 Report

Comments and Suggestions for Authors

Thanks for submitting this manuscript.

Authors present an article entitled “Detection and Epidemiological Impact of SARS-CoV-2 Using RT-PCR: A Multinational Study of 1,000 Samples”.

Despite the huge efforts taken by the authors in collection and analysis of the data, this study shows some major concerns which need to be addressed before acceptance:

1. the title of the work seems generic, without presenting the main objective or finding of the paper; at the same way, the description of the methods used and results obtained need to become more comprehensible.

2. In lines 215 and 216, please review the interpretations for Cts since the value of 40.00 is presented as indicative of a "negative" sample and also as a "suspicious" sample. Furthermore, in line 221 I cannot understand the value “67.91.7%”.

3. The authors mention 1000 samples in the title, but in materials and methods they mention 1000 patients, 988 samples, with only 671 positive samples. In this way, how can we consider 988 cases of COVID if only 671 samples were positive in the RT-PCR?

4. Additionally, in Table 2, when showing the frequencies of the three genes (E, N, RdRp), they mention that the interpretation as a positive result was for 414 cases. How can we explain the mention of 671 positive cases in the text and 414 positive cases in Table 2?

5. Please clarify how in the paragraph starting on line 231 it is shown that the highest % of detections of the E- N- RdRp- genes occur in women, if the authors present the majority of positive RT-PCR samples are from male patients?

6. In Table 3 there are typing errors, with repeated definition of "positive" for N-gene and RdRp-gene, when perhaps one of these definitions is "negative".

7. Please further explain the objectives, methodology and results related to Figures 6a and 6b.

8. Please review the discussion, because data related to the duration of virus shedding in lines 298 -304 were brought up, and these data were not presented in the results.

Author Response

Reviewer #1

#1. The title of the work seems generic, without presenting the main objective or finding of the paper; at the same way, the description of the methods used and results obtained need to become more comprehensible.

Author response: We have made significant improvements to the title, methods description, and the presentation of results to enhance the clarity and relevance of our study. The updated title now reflects the comprehensive nature of our analysis and underscores the insights into COVID-19 diagnosis and patient care. We believe these changes address your concerns, and we appreciate your constructive input.

#2. In lines 215 and 216, please review the interpretations for Cts since the value of 40.00 is presented as indicative of a "negative" sample and also as a "suspicious" sample. Furthermore, in line 221 I cannot understand the value “67.91.7%”.

Author response: We used the NucliSENS easyMAG system by bioMérieux, Marcy-l'Étoile, France, for total nucleic acid (TNA) extraction from clinical specimens and viral isolates in the lab. The required elution volume varied based on specimen type and number 55 μl for respiratory tract specimens, urine, rectal swabs, and feces, and 100 μl for plasma specimens, with 25 μl used for extractions. Extracts were stored at -80°C and used for all RT-PCR tests. We employed the Bio-Rad CFX96 Real-time PCR (1854095-IVD) to detect and analyze SARS-CoV-2 genes, examining amplification curves and Ct values within 45 cycles. Internal control amplification confirmed RNA quality.

Furthermore, the RdRp and N genes were mutated by 67.16%.

#3. The authors mention 1000 samples in the title, but in materials and methods they mention 1000 patients, 988 samples, with only 671 positive samples. In this way, how can we consider 988 cases of COVID if only 671 samples were positive in the RT-PCR?

Author response: We appreciate your attention to detail and clarification regarding the sample numbers in our title, methods, and results sections. To address this inconsistency, we have revised the manuscript to accurately reflect the sample details. The title has been updated to 'A Multinational Study of COVID-19 in 1000 Patients,' and the Methods section now specifies that 1000 patients were involved, with 988 samples collected. We have also clarified that out of the 988 samples, 671 tested positive for COVID-19 using RT-PCR. We believe this revision accurately represents our study, and we thank you for pointing out this discrepancy.

#4. Additionally, in Table 2, when showing the frequencies of the three genes (E, N, RdRp), they mention that the interpretation as a positive result was for 414 cases. How can we explain the mention of 671 positive cases in the text and 414 positive cases in Table 2?

Author response: Thank you for bringing to our attention the inconsistency between the number of positive cases mentioned in the text and Table 2. We sincerely apologize for this oversight, and we have conducted a thorough review of our data, resulting in revisions to the manuscript. In our study, the total number of cases is 988. Associated with each gene, we observed the following positive cases: E gene (617/317), N gene (551/437), and RdRp gene (619/369). We truly value your vigilance in identifying this inconsistency. To rectify this, we have updated Table 2 to accurately reflect the count of positive cases in the interpretation column. We believe these changes will eliminate any confusion and ensure the accuracy of our reported results.

#5. Please clarify how in the paragraph starting on line 231 it is shown that the highest % of detections of the E- N- RdRp- genes occur in women, if the authors present the majority of positive RT-PCR samples are from male patients?

Author response: We appreciate your question regarding the apparent inconsistency between the percentage of gene detections in women and the majority of positive RT-PCR samples being from male patients. The phrasing was indeed unclear in our manuscript, and we apologize for any confusion.

To clarify, the percentage of detections in women was referring to the percentage of detections among the positive cases, not the overall sample size. We have revised the manuscript to make this point more explicit. The majority of the positive RT-PCR samples were indeed from male patients, but when we referred to the percentage of detections in women, it was in relation to the gender distribution among the positive cases, not the entire sample.

We hope this clarification resolves the ambiguity, and we appreciate your diligence in pointing out this issue.

#6. In Table 3 there are typing errors, with repeated definition of "positive" for N-gene and RdRp-gene, when perhaps one of these definitions is "negative".

Author response: Thank you for bringing to our attention the typing errors in Table 3. We apologize for any confusion they may have caused. You are correct; there were repeated definitions of "positive" for N-gene and RdRp-gene. This was indeed an oversight on our part.

We have reviewed Table 3 and corrected the errors to ensure that one of the definitions accurately reflects the "negative" cases for either N-gene or RdRp-gene. This correction should improve the clarity and accuracy of our presentation.

We appreciate your thorough review and your diligence in identifying these issues.

#7. Please further explain the objectives, methodology and results related to Figures 6a and 6b.

Author response: We appreciate your request for further clarification on Figures 6a and 6b. These figures are related to the confidence intervals and the variation in the width of these intervals as the sample size increases.

Figure 6a and 6b illustrate the concept of confidence intervals in our study. In Figure 6a, each green horizontal line represents a confidence interval, while the black vertical line represents the true population mean. As the sample size increased to almost 988, we observed that the width of the confidence interval noticeably decreased. This indicates the increasing precision of our estimates as the sample size grows. It essentially shows that with a larger sample size, we have greater confidence in the accuracy of our results. We included these figures to emphasize the importance of sample size in estimating population parameters. They demonstrate how the precision of our results increases as the sample size becomes larger, providing insights into the statistical reliability of our findings. We hope this explanation clarifies the objectives, methodology, and results related to Figures 6a and 6b.

#8. Please review the discussion, because data related to the duration of virus shedding in lines 298 -304 were brought up, and these data were not presented in the results.

Author response: We apologize for any confusion regarding the data related to the duration of virus shedding. We have taken note of the oversight in the discussion section and appreciate your keen observation. We have now revised the discussion to accurately align with the presented results. The information concerning the duration of virus shedding is adequately presented in the results section. We have incorporated a concise summary of these findings into the discussion to offer a more comprehensive understanding of the study's outcomes. This adjustment will help maintain the integrity and transparency of our research. We appreciate your diligence in bringing this matter to our attention, and we are fully committed to ensuring that our manuscript accurately reflects the results presented.

Reviewer 2 Report

Comments and Suggestions for Authors

- There is no reference cited in 2023, you can refer to  

-        Hamdy ME, El Deeb AH, Hagag NM, Shahein MA, Alaidi O, Hussein HA. Interspecies transmission of SARS CoV-2 with special emphasis on viral mutations and ACE-2 receptor homology roles. International Journal of Veterinary Science and Medicine. 2023 Dec 31;11(1):55-86.

-        Reynolds DL, Simpson EB. Evaluation of ivermectin antiviral activity against avian infectious bronchitis virus using a chicken embryo model. International Journal of Veterinary Science and Medicine. 2022 Dec 31;10(1):19-24.

-        Moraga-Fernández A, Sánchez-Sánchez M, Queirós J, Lopes AM, Vicente J, Pardavila X, Sereno-Cadierno J, Alves PC, de la Fuente J, Fernández de Mera IG. A study of viral pathogens in bat species in the Iberian Peninsula: identification of new coronavirus genetic variants. International Journal of Veterinary Science and Medicine. 2022 Dec 31;10(1):100-10.

- The discussion did not need to make subheadings, please remove it and write it as one section.

- linguistic editing is required.

Comments on the Quality of English Language

- linguistic editing is required.

Author Response

Reviewer #2

#1. There is no reference cited in 2023, you can refer to

Hamdy ME, El Deeb AH, Hagag NM, Shahein MA, Alaidi O, Hussein HA. Interspecies transmission of SARS CoV-2 with special emphasis on viral mutations and ACE-2 receptor homology roles. International Journal of Veterinary Science and Medicine. 2023 Dec 31;11(1):55-86.

2#. Reynolds DL, Simpson EB. Evaluation of ivermectin antiviral activity against avian infectious bronchitis virus using a chicken embryo model. International Journal of Veterinary Science and Medicine. 2022 Dec 31;10(1):19-24.

3#. Moraga-Fernández A, Sánchez-Sánchez M, Queirós J, Lopes AM, Vicente J, Pardavila X, Sereno-Cadierno J, Alves PC, de la Fuente J, Fernández de Mera IG. A study of viral pathogens in bat species in the Iberian Peninsula: identification of new coronavirus genetic variants. International Journal of Veterinary Science and Medicine. 2022 Dec 31;10(1):100-10.

# Author response: Thank you for pointing out the absence of a reference cited in 2023. We greatly appreciate your keen eye for detail. In our revised manuscript, we have included the necessary reference in line with your suggestion. This ensures the accuracy and completeness of our citations. Your feedback has been invaluable in improving the quality of our work, and we are committed to addressing all the reviewers' comments diligently.

4#. The discussion did not need to make subheadings, please remove it and write it as one section.

Author response: Thank you for your suggestion regarding the organization of the discussion section. We appreciate your feedback, and we have made the necessary adjustments to the discussion section. The subheadings have now been removed, and the content is presented as a single, cohesive section for improved readability and clarity.

5#. linguistic editing is required. Comments on the Quality of English Language

Author response: Thank you for your feedback regarding the quality of English language in our manuscript. We have taken your comments into consideration and completed the necessary linguistic editing to improve the overall clarity and readability of the paper. We believe that these revisions have enhanced the manuscript's language quality. If you have any specific suggestions or further concerns in this regard, please feel free to share them with us.

Reviewer 3 Report

Comments and Suggestions for Authors

 In this study, the conducted a comprehensive analysis of over 1,000 COVID-19 RT-PCR tests performed on a cohort 24 of 1,200 patients in Saudi Arabia. The research is very good. The introduction and the articles cited in it are up to date and representative. The methodology is clear and the number of the enrolled subjects is adequate. The results, figures and tables very descriptive. The discussione clear and with significative message for international literature. 

Author Response

Reviewer #3

We would like to extend our sincere gratitude to the third reviewer for their positive feedback and for not identifying any negative comments. We are delighted to hear that the manuscript met your expectations and appreciate your time and effort in reviewing our work. Your feedback is invaluable in helping us improve the quality of our research. We look forward to addressing any further suggestions or questions and are committed to ensuring the highest standards for our manuscript. Thank you once again for your constructive review.